# Night-Time Exposure to Road, Railway, Aircraft, and Recreational Noise Is Associated with Hypnotic Psychotropic Drug Dispensing for Chronic Insomnia in the Paris Metropolitan Area

**DOI:** 10.3390/ijerph22111647

**Published:** 2025-10-30

**Authors:** Maxime Chauvineau, Sabine Host, Khadim Ndiaye, Matthieu Sineau, Victor Decourt, Manuel Hellot, Fanny Mietlicki, Damien Léger

**Affiliations:** 1VIFASOM (UMR Vigilance Fatigue Sommeil et Santé Publique), Université Paris Cité, 75006 Paris, France; chauvineau.maxime.pro@gmail.com; 2Bruitparif, 93200 Saint-Denis, France; matthieu.sineau@bruitparif.fr (M.S.); victor@bruitparif.fr (V.D.); manuel.hellot@bruitparif.fr (M.H.); fanny.mietlicki@bruitparif.fr (F.M.); 3Observatoire Régional de Santé (ORS) Île-de-France, 93200 Saint-Denis, France; sabine.host@institutparisregion.fr (S.H.); khadim.ndiaye@institutparisregion.fr (K.N.); 4Assistance Publique–Hôpitaux de Paris (AP-HP), Hôtel-Dieu, Centre du Sommeil et de la Vigilance, CRPPE Sommeil Vigilance et Travail, 75004 Paris, France

**Keywords:** noise pollution, sleep disorders, insomnia, hypnotics, recreational noise, public health

## Abstract

Urban environmental noise represents a major public health issue contributing to chronic sleep disturbances, mainly from road, aircraft, and railway traffic. Night-time recreational noise from cafés, bars, and restaurants has emerged as a frequent source of sleep complaints but remains poorly understood, along with the influence of sociodemographic and economic factors. We addressed this gap by conducting a large-scale ecological study across the Paris Metropolitan Area (~10.5 million inhabitants) examining associations between the Average Energetic Index of night-time noise (AEI L_n_) from road, aircraft, railway, and recreational sources and the prevalence of adults aged 18–79 reimbursed for hypnotic psychotropic drugs prescribed for chronic insomnia between 2017 and 2019, stratified by sex, age, and socioeconomic status. The AEI L_n_ represents the population-weighted average energy noise level within each territory at night (22:00–06:00 in France), calculated at the IRIS level (~2487 inhabitants per IRIS). The dispensing of hypnotic psychotropic drugs concerned 513,276 inhabitants (65.4 per 1000 inhabitants [‰]) on average per year. About 8 million inhabitants (75.7%) are exposed to night-time road traffic noise exceeding WHO health guidelines, followed by railway (~1.2 million, 11.6%), recreational (~1.2 million, 11.5%), and aircraft noise (~1.0 million, 9.8%). Each 5 dB(A) increase in AEI L_n_ was significantly associated (*p* < 0.001) with higher dispensing of hypnotic psychotropic drugs, strongest for road noise (+1.0‰), followed by recreational (+0.8‰), aircraft (+0.5‰), and railway noise (+0.3‰). Effects were significantly greater among individuals aged >50 (+8.3‰), women (+2.0‰), and those in the most socioeconomically deprived areas (+2.5‰). These results support proactive public policies aimed at reducing noise from transportation and recreational activities in densely urban areas to mitigate chronic insomnia.

## 1. Introduction

Environmental noise is a growing public health and societal concern, as it is one of the most significant contributors to the deterioration of quality of life in urban populations [1]. According to the noise level recommendations of the World Health Organization (WHO) for the European Region [2], the latest estimates from the European Environment Agency (2025) show that up to 30% of Europeans (~150 million people) are exposed to long-term noise levels from traffic (i.e., road, rail, and aircraft) that are harmful to their health [3]. Road traffic remains the most widespread source in the urban setting, with ~116 million people exposed to harmful day–evening–night noise levels (≥53 dB), followed by railway noise (19.3 million ≥54 dB), and aircraft noise (14.6 million ≥45 dB). These health burdens include annoyance, sleep disturbance, cardiovascular and metabolic diseases, mental health disorders, and premature death. This represents a burden of disease amounting to 1.72 million healthy life years lost annually in Europe, associated with estimated annual economic costs of EUR 120.6 billion [3].

Chronic sleep disturbances represent the main health burden associated with environmental noise, with ~4.6 million Europeans affected by severe long-term sleep disturbance due to traffic noise [3,4]. Several experimental and epidemiological studies have shown that exposure to night-time traffic noise impairs both self-reported and objective measures of sleep quantity and quality, causing difficulties in falling asleep, reduced total sleep time, and alters sleep architecture through intermittent and premature awakenings as well as shifts to lighter, less restorative sleep stages [5,6,7,8,9,10,11]. These alterations arise through activation of the brain’s alert systems in response to noise (direct pathway), which triggers stress responses (e.g., increased blood pressure and heart rate, body movements) via stimulation of the sympathetic nervous system and hypothalamic–pituitary–adrenal axis, resulting in the release of stress hormones such as cortisol, adrenaline, and noradrenaline [12,13]. The indirect pathway is mediated by cognitive perception of noise, including annoyance and noise sensitivity [14,15]. A long-term exposure to night-time traffic noise keeps the body in a constant state of arousal, which may contribute to chronic sleep disturbances and circadian rhythm disruptions [13,16], particularly promoting chronic insomnia symptoms (i.e., difficulties falling asleep, awakenings during the night, and waking up too early) [17]. The negative effects of insufficient and disrupted sleep on health and daytime functioning are well established and include increased daytime fatigue, sleepiness, mood disorders, reduced concentration, learning difficulties, and decreased cognitive performance [18,19]. Inadequate sleep may also impair hormonal regulation, glucose metabolism and blood pressure control, which increased risk of cardiovascular disease, insulin resistance, obesity, and type 2 diabetes [13,16,20]. The pervasive presence of traffic noise and chronic exposure in industrialized countries make traffic noise-related chronic sleep disturbance a major public health issue.

Beyond traffic, evening and night-time recreational noise has recently emerged as a major source of environmental noise pollution in European cities with strong tourist, cultural, and festive appeal, as city centres are increasingly transformed into night-time leisure zones [21]. With the expansion of the night-time economy, along with societal changes such as the spread of outdoor terraces, public drinking, and the development of portable sound systems, more and more people gather to enjoy public spaces and establishment terraces, which has intensified conflicts between bar and music venue owners, night-time revellers, and residents, mainly due to noise annoyance. For example, recent changes in urban space use and commercial activity in Paris (+4.5% increase in bars and restaurants between 2017 and 2020 [22]) and the proliferation of terraces following the health crisis caused by COVID-19 [23] have provoked strong reactions from residents regarding noise. In 2021, 11% of Parisians cited recreational activities as the main source of noise annoyance in their homes [24], prompting growing calls for authorities to address night-time noise and sleep disturbances, a concern heightened by their potential long-term health effects [25]. However, at present, the phenomenon remains poorly investigated. For instance, recreational noise is not considered in the European Environment Agency’s reports on environmental noise in Europe [3,26]. To the best of our knowledge, only one study has examined noise from recreational activities in relation to non-auditory health outcomes, finding a significant association with increased hypertension risk, but without assessing its effects on sleep disturbances [27]. Furthermore, no study has examined the respective impact of transportation noise sources (road, aircraft, and railway), along with recreational noise, on chronic sleep disturbances in a large-scale field study.

Finally, current scientific knowledge on the effects of noise on sleep is largely based on studies conducted under controlled laboratory conditions (collecting objective sleep parameters) in small samples, and on questionnaire-based surveys (collecting self-reported sleep disturbances), which may be influenced by question wording and personal interest [28]. To overcome these limitations, data on reimbursements for medications used to treat sleep disorders offer an objective and scalable indicator for analysing chronic sleep disturbances in large-scale field studies. This measure has been suggested to be more reliable than self-reported information on sleep disturbances, which may be more prone to information bias [29]. Few studies have examined the relationship between environmental noise exposure and registry-based prescription of medication to treat sleep disorders, yielding contradictory results [30,31,32]. Existing studies are limited, focusing only on certain drugs (e.g., benzodiazepines or hypnotics) and not estimating the prevalence of individuals with chronic sleep disorders based on prescription chronicity and exclusion criteria related to use for other pathologies. Moreover, the influence of sociodemographic and economic factors (e.g., sex, age, socioeconomic status, population density) related to the impact of environmental noise on sleep medication use [33,34,35] has not always been considered, particularly at a fine spatial resolution.

The aim of the present study was to examine whether individual and combined exposures to road, aircraft, railway, and recreational noise sources (from cafés, bars, and restaurants) were associated with registry data on reimbursements for hypnotic psychotropic drugs prescribed for chronic insomnia-type sleep disorders, in a large-scale ecological study conducted at the census block level in the Paris Metropolitan Area (the most densely populated and urbanized area of France) during the years 2017 to 2019. We also aimed to explore the potential association of these data with sociodemographic and economic factors, including sex, age, and the socioeconomic status of the place of residence.

## 2. Materials and Method

### 2.1. Study Setting and Area

We focused on 432 municipalities and 20 Parisian arrondissements comprising the Métropole du Grand Paris and 13 surrounding agglomeration communities in the Île-de-France region (Figure 1). The area covered 3007 km^2^ and had a total population of 10.5 million in 2016 (data source: French National Institute for Statistics and Economic Studies, INSEE, 2016). It was included in the list of agglomerations covered by the European Environmental Noise Directive 2002/49/EC [36] on environmental noise [37]. All data were extracted for the period 2017–2019. Analysis of hypnotic psychotropic drug dispensing was conducted at the level of municipalities and Parisian arrondissements. Environmental noise exposure data, available at a finer spatial resolution, along with demographic and socio-economic status information, were analysed by subdividing the study area into 4229 census blocks (called IRIS by INSEE), each comprising ~2487 inhabitants on average.

### 2.2. Hypnotic Psychotropic Drug Database

#### 2.2.1. Data Sources

Information on the dispensing of hypnotic psychotropic drugs was extracted from the French National Inter-Regime Health Insurance Information System (SNIIRAM, SNDS, in French, Système national des données de santé) [38]. The SNIIRAM is a pseudonymized database containing extensive information on drug reimbursements for almost the entire French population as health insurance is mandatory in France. The Inter-Regime Consumption Datamart (DCIR) database was used to access comprehensive individual-level data on the dates and detailed coding of drugs dispensed at community pharmacies, along with patients’ personal details (date of birth, sex and place of residence). We followed the reference methodology MR-007 of the Commission nationale de l’informatique et des libertés (CNIL), which provides a framework for accessing data from the SNDS and conducting retrospective studies [39].

#### 2.2.2. Selection of Drugs

A list of 381 drugs, reimbursed by the French health insurance system, was retained by a senior sleep medicine specialist (DL) as proxy for chronic insomnia-type sleep disorders. It included strictly hypnotic drugs (i.e., substances capable of inducing and/or maintaining sleep) and psychotropic drugs commonly used as sedatives at night to treat insomnia, even though they do not have a primary hypnotic indication. These drugs had marketing authorisation granted by the French Medicines Agency (Agence nationale de sécurité du médicament et des produits de santé, ANSM) and belonged to the following Anatomical Therapeutic Chemical (ATC) classification groups: N05 (psycholeptics, *n* = 276), N06 (psychoanaleptics, *n* = 87), R06 (antihistamines for systemic use, *n* = 6), V03AX (other therapeutic products, *n* = 9) and Z (no ATC code assigned, *n* = 3). Details on drugs grouped by the ATC classification are presented in Appendix A. The selection was made based on the list of originator and generic-name drugs in the Vidal database and cross-referenced with Code Identifiant de Présentation (CIP) codes derived from drug reimbursement data in France, available in the SNDS.

#### 2.2.3. Patients Identification

Figure 2 presents the flowchart for patient identification. It was conducted using sliding three-month periods to capture the consecutive and progressive nature of the treatment. For each of these periods, patients were included if they received a prescription for at least one drug belonging to the selected pharmaceuticals covering ≥3 consecutive months of treatment. They were also included only if aged between 18 and 79. The rationale for this criterion was that few sleep treatments are prescribed for individuals under 18, and because the high prevalence of sleep disorders in the elderly is largely driven by age itself [40], which may outweigh the effect of environmental noise exposure. The number of medication boxes prescribed for treatments lasting ≥3 months was used to define treatment chronicity. Patients with psychiatric disorders were excluded, as the drugs on the established list are often used by this population, even when not prescribed for sleep disorders [41]. They were identified through the methodology of the French National Health Insurance Fund (CNAM), which algorithmically classified Individuals into 58 groups of pathologies among them psychiatric pathologies. Patients belonging to the following subgroups were excluded: “psychotic disorders”, “mental deficiencies”, “addictive disorders”, “psychiatric disorders that began in childhood”, “other psychiatric disorders”, and those reimbursed for “antidepressant, lithium, depakote, and depamide treatments (excluding pathologies)”, whereas patients with “neurotic and mood disorders” were included. Indeed, these conditions, in the absence of a declared psychiatric illness, may be linked to anxiety potentially caused by exposure to environmental noise [42]. Patients reimbursed for benzodiazepine-derived anxiolytic psycholeptics (N05BA) and/or antidepressant psychoanaleptics (N06A) at a dosage > 1/day were excluded, as this frequency may indicate use for conditions other than chronic insomnia. The number of patients identified was calculated for each municipality and Parisian arrondissements to estimate the prevalence of patients reimbursed for hypnotic psychotropic drugs per 1000 inhabitants.

### 2.3. Night-Time Environmental Noise Exposure

#### 2.3.1. Modelling Transportation Noise

Noise levels from road, rail and aircraft traffic were estimated using the 4th round noise maps produced and published in 2023 under the framework of the European Environmental Noise Directive (2002/49/EC) [36], which requires public authorities to produce noise maps every 5 years for large urban areas (≥100,000 inhabitants) and major transport infrastructures. Noise levels were modelled according to the European CNOSSOS-EU methodology [43] and the Commission Directive (EU) 2015/996 [44], which determine the methods for calculating noise emission levels, the effects related to the propagation of noise in the environment, the methodology for positioning receptors for calculating noise levels and the method for assigning populations to these receptors.

Briefly, the process consisted of developing a numerical model and then performing acoustic simulations to estimate noise levels from transport sources at receiver points located at a height of 4 m above the ground on each building façade, with a horizontal resolution of 5 m or less. In addition to sound sources, the model considers input related to topography (terrain), buildings, ground effects, and acoustic barriers (for ground transportation noise), as well as the effect of long-term weather conditions (temperature, relative humidity, wind speed, and direction) on noise propagation and reflection to assess ambient noise levels throughout the study area (see Appendix A for further details).

Transportation noise maps were produced by Bruitparif based on input and traffic data collected between 2017 and 2019, originating from various stakeholders, including local authorities, infrastructure managers, and flat-rate data. They incorporated the contributions required for major transport infrastructures under the European Environmental Noise Directive (2002/49/EC) [36]:CEREMA, for road and rail traffic in areas located outside the Métropole du Grand Paris, covering road infrastructure with >3 million vehicles per year and railways with >30,000 trains per year;ADP (Paris Airports) and the DGAC (French Civil Aviation Authority), for aircraft traffic at major airports (>50,000 movements per year).

These contributions were complemented by Bruitparif across the study area by modelling transportation noise source emissions and propagation using CadnaA software (version 2022 MR2: build 193.5260, DataKustik, Greifenberg, Germany). Modelling noise propagation and reflection required a digital terrain model obtained from the IGN (National Geographic Institute)’s 2019 BD TOPO^®^ [45]. It included shapefiles of buildings (polygons with attributes such as usage, height, and number of dwellings), topography, roads and railway tracks (polylines), and other elements such as administrative boundaries and natural areas.

Transportation noise emissions were calculated from average annual traffic conditions, including traffic flow, speeds, traffic composition (light vehicles, heavy goods vehicles, motorized two-wheelers), road surface type, types of trains, railway track equipment, average flight paths, and aircraft fleet composition. For road noise, traffic data were derived from traffic counts available in the area (permanent counters, temporary campaigns, cameras, data from navigation applications, etc.). For railway noise, the input data (traffic, train types, infrastructure characteristics) came exclusively from infrastructure managers (RATP and SNCF). For aircraft noise, calculations were performed by airport management services for air traffic control authorities (DGAC) based on annual averages in terms of flight paths and number of movements per aircraft type.

#### 2.3.2. Modelling Recreational Noise

Unlike transportation noise, there is currently no standardized methodology for the acoustic modelling of recreational noise. Therefore, a specific proxy for noise levels associated with recreational activities from cafés, bars, and restaurant establishments was developed. The process involves several steps (see Appendix A for further details):*(1)* *Identifying and geolocating recreational establishments with terraces likely to generate noise based on their activity classification*

As no database provided an exhaustive list of recreational establishments with terraces throughout the study area, identification relied on the 2022 Sirene company register [46]. A filter was applied to retain only food service activities (code 56.10A) and beverage-serving activities (code 56.30Z) according to the French classification of activities [47], and duplicate entries were removed. This database had the advantage of covering the entire study area and being regularly updated. In total, 59,110 establishments (49,709 food service and 9401 beverage-serving) were identified within the study area, including 31,817 in the city of Paris alone (27,476 food service and 4341 beverage-serving). The establishments were geolocated with Geographic Information Systeme (GIS) as point features using the X and Y coordinates from the Sirene database, projected in the Lambert-93 coordinate system.

*(2)* 
*Creating a surface element representing a terrace (flat-rate surface area) in front of the building corresponding to each identified establishment*


“Mirror points” were generated in the GIS to construct virtual terraces for each establishment, located outside the building and placed symmetrically to the original geolocation point. These surfaces were adjusted to the geometry of the nearest building façade referenced in the 2019 BD TOPO^®^ (the same database used for transportation noise) [45]. Virtual terraces were then drawn as 5-meter-radius discs centered on the mirror points, with the portions intersecting building polygons (expanded by 20 cm to avoid modelling errors in CadnaA) removed. Intersections between adjacent terraces were also eliminated. Finally, terrace geometries were simplified to reduce vertex counts and optimize acoustic modelling computation times. These processes were carried out using a Python (version 3.13.5) script using the “GeoPandas” and “shapely” libraries.

*(3)* 
*Assigning an acoustic power level to the surface*


Each virtual terrace surface was treated as a surface sound source positioned 1.10 m above ground level (corresponding to the head height of a seated person) and assigned an acoustic power level. The octave-band acoustic power levels per unit area (Lw/m^2^) attributed to the terraces were derived from the results of our study conducted in 2022 by Bruitparif in the Parisian neighbourhood of Halles–Beaubourg–Montorgueil [48]. This study determined an average acoustic power spectrum using inverse sound propagation modelling based on environmental noise measurements. This methodology, implemented using CadnaA software, involves deducing the sound power of emitting surfaces, defined by their geometry and surface area, from actual measurements taken at several receiving points. The software then automatically adjusts the source levels until the simulation best matches the measurements. Terrace sound emissions consisted of a mixture of human voices and amplified music played by the establishments, either outdoors or indoors, propagating to the exterior. In the present study, the acoustic power levels were adjusted based on the ratio between the total terrace surface area in the Parisian neighbourhood of Halles–Beaubourg–Montorgueil (10,288 m^2^) and the total terrace surface area in the present study (30,763 m^2^). This resulted in a correction factor of −5 dB for the acoustic power levels emitted by terraces, approximating the logarithmic ratio of the total terrace areas obtained using the two identification methods, calculated as $10\cdot \;\log10(\frac{10,288\;m^{2}}{30,763\;m^{2}})$. This correction involved adjusting the acoustic power levels emitted by terraces to ensure that the estimated recreational noise impact on the population remained consistent with that of the Halles–Beaubourg–Montorgueil Parisian neighbourhood [48]. An additional correction of −3 dB was applied to adjust from an assessment based on the summer period to an annual average assessment. This correction amounts to considering that terraces are in operation for about half of the year.

*(4)* 
*Simulating noise propagation into the environment*


The propagation modelling of noise generated by the terraces was performed using CadnaA software, applying the same method as for transportation noise modelling (CNOSSOS-EU methodology [43]) implemented to produce noise maps according to the European Environmental Noise Directive 2002/49/EC [36].

#### 2.3.3. Estimation of the Noise-Exposed Population

The indicator used to characterize night-time noise was the annual average L_n_ indicator, which is the regulatory indictor recommended by the European Union to produce night-time noise maps and for environmental noise management regarding sleep disturbances [3,36]. It refers to the A-weighted annual average sound pressure level during the night period (22:00–06:00 in France), expressed in dB(A). “A-weighted” means that the sound pressure levels are adjusted to consider the physical sensitivity of human hearing at different sound frequencies.

We estimated the population exposed to each noise source at night according to the CNOSSOS-EU methodology [43] and the Commission Directive (EU) 2015/996 [44]. This approach consists of assigning each building’s inhabitants to façade noise receivers: for multi-dwelling buildings, for the upper half of the noisiest façade receivers; and for single-dwelling buildings, to the most exposed façade receiver. The distribution of the population by noise exposure level was reported in 1 dB(A) intervals for road, rail, and recreational noise, and in 5 dB(A) intervals for aircraft noise. The number of inhabitants exposed to night-time noise levels exceeding the WHO recommendations [2] was then calculated for each environmental source (i.e., road: 45 dB(A), rail: 44 dB(A), aircraft: 40 dB(A)), along with those exceeding the regulatory limit values adopted by France (road: 62 dB(A), rail: 65 dB(A), aircraft: 50 dB(A)) within the framework of the transposition of the European Environmental Noise Directive 2002/49/EC [36]. In the absence of WHO recommendations and legal limit values for recreational noise in France, those for road traffic noise were applied as the most conservative benchmark. This choice is also justified as recreational noise is temporally more similar to road noise than to railway or aircraft noise, the latter being event-driven and producing roughly regular peaks. The population exposed to a less conservative night-time noise level, i.e., 40 dB(A) and 50 dB(A), was also estimated for recreational noise. The 40 dB(A) threshold has been suggested as the lowest observed adverse effect level for night noise, associated with self-reported sleep disturbance, environmental insomnia, and increased use of somnifacient drugs and sedatives [3,49].

#### 2.3.4. Estimation of Noise Levels at IRIS

A population-weighted average noise level, called average energetic index (AEI) [50,51], was estimated at IRIS level for each noise source (road, rail, aircraft and recreational) as a measure of environmental noise exposure potentially related to the local prevalence of patients reimbursed for hypnotic psychotropic drugs. It is defined as the population-weighted logarithmic mean of façade sound levels, where each façade point is weighted by the number of residents in its building divided across its façade points. The AEI L_n_ was calculated for each territorial unit (IRIS) by weighting noise levels by the number of inhabitants exposed in each 1 dB(A) interval. For an IRIS *i*, and for the L_n_ indicator, the AEI L_n_ was defined as follows:$$AEI\;L_{n}=10\cdot \log_{10}\left(\frac{1}{\sum_{k}Pk}\sum_{j\geq 1}Pj10^{\frac{L_{nj}}{10}}\right)$$
where $L_{nj}$ is the averaged energy-equivalent sound level *j* over the night period (the difference between $L_{n}$_j_ and $L_{n}$_j+1_ is equal to 1 dB(A)), Pj the number of inhabitants in the IRIS exposed to $L_{nj}$ dB(A), and $\frac{1}{\sum_{k}Pk}$ is the total number of inhabitants in the IRIS.

The AEI L_n_ was also estimated for combined transportation noise sources (road, rail and aircraft) and for all sources combined (road, rail, aircraft and recreational), calculated through the energetic summation of the noise levels from each source.

#### 2.3.5. Covariates

Age, sex, log-transformed population density, a deprivation index at IRIS level (data source: INSEE, 2020), and the proportion of the population reporting a primary care physician at the municipality level (source: SNDS, 2017–2019) were included as *a priori* confounding factors. The log-transformed population density was used instead of the raw population density to account for the large variation in density across IRISs. The proportion of the population reporting a primary care physician was included to account for potential bias related to self-medication and access to healthcare, which may influence the dispensing of hypnotic psychotropic drugs. The deprivation index, proposed by Rey et al. [52], was constructed to characterize the socioeconomic status of each IRIS based on four variables, each representing a dimension of the socioeconomic level: median household income, percentage of post-secondary study graduated ≥2 years in the population aged ≥ 15 years, percentage blue-collar workers in the active population, and unemployment rate. The deprivation index was defined as the first principal component derived from a principal component analysis of these four variables. The study area was then divided into quintiles of deprivation index (1 = least disadvantaged IRISs; 5 = most disadvantaged IRISs).

## 3. Statistical Analysis

Data processing and statistical analyses were performed using R software (version 4.4.2). The relationship between the prevalence of patients reimbursed for hypnotic psychotropic drugs and environmental noise exposure (AEI L_n_) was assessed using generalized additive mixed models (GAMMs, “gam” function from the “mgcv” package), including the IRIS as a random effect to account for data non-independence at IRIS level [53]. They were estimated using the Restricted Maximum Likelihood (REML) method.

A first GAMM was fitted including each noise source (road, aircraft and rail traffic, and recreational) as a fixed effect. A second GAMM was fitted for combined transportation noise sources (road, rail and aircraft) and for all sources combined (road, rail, aircraft and recreational). These models were adjusted for the following sociodemographic and economic confounders: deprivation index quintile (categorical variable), proportion of the population reporting a primary care physician, and log-transformed population density. They used the age-standardized prevalence of patients reimbursed for hypnotic psychotropic drugs as the outcome variable to adjust for age, which may vary across IRIS units. This age-standardization corresponds to the ratio of the observed number of patients in each IRIS to the expected number if the IRIS had the same age structure as the standard population.

Three additional GAMMs were employed to assess the main effects of sex, age, and deprivation index quintile (separately) on the prevalence of patients reimbursed for hypnotic psychotropic drugs, as well as their two-way interactions with combined exposure to all noise sources (road, rail, aircraft, and recreational). Two further GAMMs were then fitted, each including a three-way interaction—(1) sex × age × combined exposure to all noise sources, and (2) sex × deprivation index quintile × combined exposure—to assess whether the effects of age and deprivation differed between men and women under varying levels of overall noise exposure. These models included the following confounders: proportion of the population reporting a primary care physician, and log-transformed population density.

A smoothing spline function was initially included in the models to account for potential nonlinear effects. As the results suggested approximately linear relationships, the spline function was omitted. Collinearity between independent variables was assessed for each model, with all variance inflation factors (VIF) < 5, indicating low collinearity. The GAMM results are presented as estimate with 95% confidence intervals (CIs), representing the absolute change in number of patients (per 1000 inhabitants) reimbursed for hypnotic psychotropic drugs per 5 dB(A) increase in noise level. Statistical significance was set at *p* < 0.05.

## 4. Results

### 4.1. Prevalence of Chronic Dispensing of Hypnotic Psychotropic Drugs

The dispensing of hypnotic psychotropic drugs was observed in 513,276 inhabitants (65.4 per 1000 inhabitants [‰]), on average per year between 2017 and 2019. Prevalence was significantly higher in women than in men (*p* < 0.001). Mapping of the prevalence of patients reimbursed for hypnotic psychotropic drugs in the study area per sex are presented in Appendix A. It increased progressively with age and was the highest among individuals aged 65–79 years (*p* < 0.001). Prevalence significantly increased with the deprivation index and was the highest in the most socioeconomically disadvantaged areas (quintile 5; *p* < 0.001; Table 1).

### 4.2. Mapping and Population Exposed to Environmental Noise

The night-time (L_n_) IRIS-level noise exposure in the study area for each environmental noise source is presented in Figure 3. Road traffic noise is widespread across the study area. Rail traffic noise is highly concentrated along the railway tracks, while aircraft noise is primarily located under main takeoff and landing paths of Paris-Orly and Paris-Charles de Gaulle airports. Recreational noise is mainly concentrated within the central city of Paris.

Road traffic was the noise source with the highest prevalence of exposure at or above the night-time WHO guideline levels of 45 dB(A) (~8.0 million, 75.7%, Table 2). Railway noise ranked second, with ~1.2 million ≥ 44 dB(A) (11.6%), followed by recreational noise (~1.2 million ≥ 45 dB(A) [11.5%]), and aircraft noise (~1.0 million ≥40 dB(A) [9.8%]). The prevalence of exposure to night-time recreational noise levels ≥ 40 dB(A) was 1.8 million people (17.4%).

### 4.3. Relationship Between Night-Time Source-Specific Noise Exposure and Dispensing of Hypnotic Psychotropic Drugs

Table 3 presents the association between the prevalence of patients (/1000 inhabitants) reimbursed for hypnotic psychotropic drugs and night-time noise levels (AEI L_n_), as estimated from the GAMM models. Every additional 5 dB(A) in AEI L_n_ was associated with an increase in the number of patients (/1000 inhabitants) reimbursed for hypnotic psychotropic drugs, with the strongest effect observed for road traffic noise, followed by recreational noise, aircraft noise, and rail noise.

### 4.4. Influence of Sex, Age and Deprivation Index

Figure 4 presents the relationships between combined night-time environmental noise levels (road, rail, aircraft and recreational) and the prevalence of patients (/1000 inhabitants) reimbursed for hypnotic psychotropic drugs, stratified by sex, age, and deprivation index. Among women, each 5 dB(A) increase in AEI L_n_ was associated with an increased prevalence of 1.86 (95% CI: 1.77–1.95, *p* < 0.001). Among men, a significant decrease was observed (β = −0.50 [−0.62–−0.38], *p* < 0.001).

Regarding age, the largest increases in prevalence of patients per 5 dB(A) in AEI L_n_ were observed in the 65–79 year age group (β = +10.37 [10.31–10.44], *p* < 0.001), followed by the 50–64 year group (β = +6.97 [6.91–7.04], *p* < 0.001), the 35–49 year group (β = +2.47 [2.40–2.54], *p* < 0.001), and the 18–34 year group (β = +1.46 [1.45–1.48], *p* < 0.001).

For deprivation index, the strongest effects in prevalence of patients per 5 dB(A) in AEI L_n_ were found in the most socioeconomically deprived IRIS units (quintile 5, β = +2.52 [2.43–2.61], *p* < 0.001), followed by quintile 4 (β = +2.34 [2.25–2.43], *p* < 0.001), quintile 3 (β = +2.08 [1.99–2.16], *p* < 0.001), and the least deprived quintiles (quintile 2: β = +1.70 [1.61–1.79], *p* < 0.001; quintile 1: β = +1.80 [1.79–1.82], *p* < 0.001).

In more detail (see Appendix A), we observed that the prevalence of patients per 5 dB(A) increase in AEI L_n_ progressively rose across all age groups among women, whereas it increased only in the 50–64 and 65–79 age group in men. Similarly, the effect of noise exposure increased across all deprivation index quintiles among women, whereas it increased only from the most deprived quintiles (4 and 5) among men.

## 5. Discussion

Our study examined whether road, aircraft, railway, and recreational night-time noise are associated with registry-based reimbursements for hypnotic psychotropic drugs in the Paris Metropolitan Area (2017–2019 period), while accounting for sociodemographic and economic factors (sex, age, and deprivation index). We found that: (1) road traffic had the highest prevalence of exposure (75.7% of the total population exposed to >45 dB(A) of L_n_), followed by recreational, rail, and aircraft noise; (2) an increase in AEI L_n_ level, for each of the environmental noise sources (road, rail, aircraft, and recreational), was associated with a higher prevalence of hypnotic psychotropic drug dispensing, with the strongest effect observed for road traffic, followed by recreational, aircraft, and rail noise; and (3) the association between environmental noise (all sources combined) and hypnotic psychotropic drug dispensing increased with age (especially from aged >50), was more pronounced in women than in men, and was greater in the most socioeconomically deprived areas.

### 5.1. Exposure to Environmental Noise in the Paris Metropolitan Area

The first main finding was the large proportion of the population exposed to long-term road traffic noise at night, with an estimated 75.7% (~8.0 million inhabitants) at or exceeding the WHO guideline threshold of 45 dB(A) of L_n_, considered harmful to health [2]. Railway noise ranked by far second (11.6% ≥44 dB(A)), followed by recreational noise (11.5% ≥45 dB(A)) and aircraft noise (9.8% ≥40 dB(A)). The prevalence of exposure to night-time recreational noise increased to 17.4% when applying the 40 dB(A) threshold, considered the lowest observed adverse effect level for night noise, associated with self-reported sleep disturbance, environmental insomnia, and increased use of somnifacient drugs and sedatives [3,49]. These findings support the European Environment Agency that road traffic is the primary and most widespread noise source to address in the most densely populated and urbanized area to prevent noise-related public health and societal issues [3]. Maps of the L_n_ indicators show that road traffic noise is widespread across the study area due to the large number of road infrastructures distributed across the territory, with the highest levels along major roads (e.g., motorways, ring road). Aircraft noise is primarily concentrated under main flight paths along the axes of the runways on either side of the airports. Railway noise is highly concentrated along the rail tracks. Finally, recreational noise is particularly concentrated within the central city of Paris, which accounts for 54% of all mapped establishments within the study area. The distribution of the population exposed to night-time recreational noise shows two main ranges: the first, around 15 dB(A) of L_n_, corresponds to populations with little or no exposure, while the second, around 62 dB(A) of L_n_, corresponds to populations most exposed, located in the immediate vicinity of the modelled establishments and terraces.

### 5.2. Impact of Environmental Noise on Dispensing of Hypnotic Psychotropic Drugs

To the best of our knowledge, the present study is the first to investigate the effect of four urban environmental noise sources (road, rail, aircraft, and recreational) on the dispensing of medications to treat chronic insomnia-type sleep disorders in such a large population (~10.5 million inhabitants). Our study area, urban and densely populated, exhibited overlapping exposures to the different noise sources, requiring their combined analysis in our models. After adjusting for sociodemographic confounders and each noise source, we found that increases in noise levels, whether road, aircraft, rail, or recreational, were independently associated with a higher prevalence of patients reimbursed for hypnotic psychotropic drugs.

These findings are consistent with numerous epidemiological and experimental studies reporting that traffic noise (road, aircraft, and rail) affects sleep physiology, increases the prevalence of sleep disturbances, and contributes to population sleep debt [5,6,35]. Nevertheless, studies on medication use for sleep disorders show conflicting results regarding environmental noise exposure, potentially due to, at least in part, differences in data sources. For example, three European studies on road traffic noise and self-reported sleep medication found no significant associations [17,19,54], whereas several large-scale studies using reimbursement data for hypnotic and/or psychotropic drugs reported positive associations [30,31,32]. Reimbursement data from health insurance has been suggested to be more reliable than self-reported information on sleep disturbances [29], which may be more prone to recall bias or to patients’ lack of knowledge about their medications [55]. The present comprehensive data on reimbursed drugs, including strictly hypnotic and psychotropic drugs commonly used as sedatives at night to treat insomnia, at a large scale and across a wide population, has likely provided an accurate estimate of the prevalence of chronic insomnia and of the impact of various environmental noise sources on sleep.

We observed differences in the magnitude of the effect of environmental noise on the dispensing of hypnotic psychotropic drugs across sources. Night-time road traffic noise showed the strongest significant positive association, with an estimated increase of 1.01 patients per 1000 inhabitants reimbursed for hypnotic psychotropic drugs for each additional 5 dB(A) in AEI L_n_. Combined with the fact that this source is by far the main environmental noise nuisance in our study area and in European urban environments [3], these findings suggest that road traffic noise at night is the most concerning source affecting sleep in urban populations. As a guide, our model estimates an average 2.47% reduction in reimbursements for hypnotic psychotropic drugs if night-time road noise alone does not exceed the WHO recommendation of 45 dB(A) across the study area, accounting for 92% of the reduction attributable to transport-related noise and 84% of the reduction attributable to all sources combined (road, rail, aircraft, and recreational). This reduction would be 1.06% if road noise is reduced by 3 dB(A)—roughly halving noise emissions—in areas exceeding 45 dB(A), i.e., 83% of the reduction from transport noise and 73% of that from all sources combined.

Road traffic noise is typically continuous or quasi-continuous, present both day and night, with peaks during rush hours. Although some individuals may develop partial habituation to the noise, this remains incomplete, as even in the absence of perceived annoyance, noise induces autonomic nervous system reactions that can disrupt sleep initiation and structure [8,42]. It has been shown that both the sleep onset phase, when evening traffic remains heavy, and the end of the night, when traffic resumes and sleep pressure has dissipated, are particularly vulnerable to road noise, as sleep is lightest during these periods [17].

The effect of night-time aircraft and railway noise on the prevalence of dispensing hypnotic psychotropic drugs was ~50% and ~65% lower than with road traffic noise, respectively. Literature findings on the relative impact of transportation noise sources on sleep disturbances are inconsistent. Basner et al. (2011) reported a stronger effect for road traffic on sleep structure and continuity, while aircraft and railway noise had a greater impact on subjective sleep quality [8]. Another study showed higher awakening probabilities for railway noise, followed by road and then aircraft noise [9], whereas, for the same sound level, aircraft noise was perceived as the most disruptive to sleep, followed by railway and road traffic noise [5,6]. In our study, the stronger impact of road traffic noise may be explained by the fact that the share of the population exposed at night to potentially harmful levels of aircraft or railway noise was more limited than road traffic noise. This may also result from the timing of exposure. Our study area included two major airports, Paris-Orly and Paris-Charles de Gaulle, the former having the distinctive feature of a night curfew from 23:30 to 06:00, which is relatively rare for an international airport in Europe. Furthermore, aircraft and railway noise have spectral and temporal characteristics that differ from road traffic noise, being composed of short but intense noise events, often rich in low frequencies and, in the case of railway noise, potentially generating vibrations [8]. The L_n_ indicator used in our study, which correspond to time-averaged energy metrics, may be better suited to assessing the impact of road traffic noise on chronic insomnia than that of aircraft or railway noise. Further research is needed to assess the impact of aircraft and railway noise on chronic sleep disorders using environmental noise exposure metrics that better account for their event-based nature.

The originality of our study lies in the inclusion of recreational noise from cafés, bars and restaurants establishments to estimate the prevalence of chronic insomnia related to this environmental nuisance. In European cities with strong tourist, cultural and festive appeal, evening and night-time recreational activities are an important source of annoyance for residents [21], especially in Paris where the recent proliferation of establishments with terraces following the health crisis caused by COVID-19 [23] have provoked strong reactions from residents regarding noise [24]. To our knowledge, only one previous study has examined recreational noise, focusing on its effects on blood pressure and hypertension risk, not on sleep disorders, and reported a significant association between higher recreational noise exposure and increased hypertension risk [27]. The present findings shows that night-time recreational noise showed the second-strongest positive relationship, after road traffic, with the prevalence of patients reimbursed for hypnotic psychotropic drugs. Recreational noise (mainly from voices and music) occurs primarily in the evening (18:00–22:00) and early night, with permanent terraces operating until 02:00 and temporary summer terraces until 23:00, while customer noise may persist on the street beyond closing hours. Our study conducted by Bruitparif in 2022 in the Parisian neighbourhood of Halles–Beaubourg–Montorgueil [48] showed that recreational noise tends to increase through the evening, peak around midnight, and then gradually decline as patrons leave. This type of noise might be particularly detrimental to sleep due to its event-based and intermittent nature, its timing in the evening and early night, and its concentration at weekends, interfering with typical rest and the onset of sleep. Given that recreational noise is particularly concentrated within the city of Paris and has not yet been thoroughly investigated, our findings suggest that local policies should place greater emphasis on addressing this form of noise pollution and its potential impact on public health.

### 5.3. Influence of Sex, Age, and Deprivation Index

Our study supports the hypothesis that certain populations may be at higher risk of chronic sleep disturbances due to environmental noise exposure [12,18]. We observed that the impact of environmental noise on the dispensing of hypnotic psychotropic drugs increases considerably with age. In particular, this increase becomes more pronounced from the 50–64 and 65–79 age groups onward compared to younger groups. It is well established that the ability to maintain sleep, total sleep duration, and the proportion of slow-wave sleep decrease with age, while the use of sleep medication increases [56,57]. Insomnia is also particularly prevalent among older adults (≥65 years) [40,58]. These alterations in sleep are attributable to a gradual weakening of circadian rhythms and homeostatic sleep regulation, as well as to changes in neuroendocrine functions involved in sleep regulation. Notably, it is primarily during slow-wave sleep that the brain filters external noise to protect the sleeper from environmental disturbances [59,60]. It is therefore plausible that lighter and more fragmented sleep increases noise sensitivity in older individuals, making them more vulnerable to the detrimental effects of environmental noise on sleep. Furthermore, studies examining perceived annoyance from environmental noise across age groups have yielded contradictory results, with some suggesting a decline in reported annoyance beyond a certain age [61]. This indicates that the greater impact of noise on sleep with age is unlikely to result from increased perception or intolerance to noise, but rather from an increased vulnerability of sleep itself.

Although less pronounced than the effect with age, the present study revealed a significant positive association between environmental noise and the dispensing of hypnotic psychotropic drugs especially in women. In detail, the effect of noise exposure increased across all age groups among women, whereas it increased only from the older age groups (50–64 and 65–79) in men. This may also be explained by a greater vulnerability of sleep among women compared with men, and by the fact that men only reach a similar level of vulnerability at more advanced ages. Numerous studies indicate that women report more sleep disturbances than men, particularly insomnia, and are more likely to use medication to address sleep problems [58]. This difference is partly explained by hormonal factors and a greater sensitivity of women to psychosocial factors related to sleep disturbances, particularly anxiety disorders [62]. For example, Halonen et al. (2012) showed that individuals with high levels of anxiety exhibited more severe insomnia symptoms when night-time noise levels exceeded 50 dB(A) [63]. Another explanation could be that women tend to have an advanced biological clock and sleep–wake rhythms compared with men [64], resulting in generally earlier bedtimes [62]. They may therefore be more exposed to evening noise, which could promote difficulties falling asleep and disturb the first part of the night. Nevertheless, Roswall et al. (2020) reported a finding contradictory to ours in a population aged 50–64 years, showing a strong positive association between road traffic noise exposure and the prescription of medications for sleep disorders in men, but not in women [30]. However, this study focused solely on road traffic noise, making comparisons with our study difficult.

Finally, our results showed strongest effects of environmental noise on hypnotic psychotropic drug dispensing in the most socioeconomically deprived areas. This effect increased across all deprivation index quintiles among women, whereas it increased only from the most deprived quintiles (4 and 5) among men, reinforcing the hypothesis that women’s sleep is more vulnerable than men’s. The first possible explanation is a higher prevalence of collective housing in the most deprived areas, which may be less well insulated against noise, thereby increasing exposure. It is also possible that more disadvantaged populations are less aware of the harmful effects of noise on health, have fewer negative perceptions and lower concern regarding environmental noise management in urban areas, and are therefore potentially less likely to adopt individual strategies to reduce their noise exposure [65,66]. Furthermore, it has been suggested that residents of the most socioeconomically deprived areas are more likely to be affected by chronic sleep disorders [67]. These populations may therefore accumulate additional risk factors for sleep disturbances, and environmental noise could further exacerbate these risks by affecting sleep that is already more vulnerable compared with that of more advantaged populations. To our knowledge, only one previous study has investigated the relationship between road traffic noise and the use of anxiolytic and hypnotic medications in urban settings according to socioeconomic status. Contrary to our findings, that study reported an increase in the dispensing of these medications only in the least deprived areas when night-time road traffic noise exceeded 55 dB(A) [31]. However, comparing our results with this study is difficult, as these medications may have been prescribed for indications other than sleep disorders, and the criteria for selecting medications and using them to identify patients with chronic sleep disorders were less stringent.

## 6. Limitations

Some limitations need to be considered when interpreting the present findings. The proxy for chronic insomnia was based solely on reimbursed prescriptions, excluding over the counter, optional, non-reimbursed, or off-label drugs, as well as information on actual use or clinical context. It therefore captures only chronic, and likely more severe, sleep disorders requiring a medical consultation and prescription, which ensures specificity but limits sensitivity and generalizability to non-clinical or acute forms. Another methodological limitation is that some patients may use these treatments for other reasons or in combination with their sleep disorders, for instance for recreational purposes [68]. We do not have data on this, but we believe that this objective measurement provides a more accurate estimate than simple questionnaires of insomnia prevalence and its association with the sound environment.

Furthermore, consensual experts recommend Cognitive Behavioural Therapy for insomnia (CBTi) rather than hypnotics as a first-line treatment to face chronic insomnia [69]. However, CBTi is often not readily available in the general population and not officially reimbursed by social security [70]. As a result, data on behavioural treatment for insomnia are not readily available and probably represent a very small proportion of the treatment of people with chronic sleep disorders. Meanwhile, medication remains readily available and inexpensive.

Another limitation of the proxy for chronic insomnia was its construction at the municipality level, preventing capture of local variability and potentially introducing ecological bias. To reduce this, noise exposure and deprivation indicators were estimated at the finer IRIS level, with IRIS-level random effects in models.

Additionally, the L_n_ indicator from noise maps reflects average exposure and cannot capture temporal patterns or specific intermittent events, particularly from air and rail traffic, and certain road traffic sources (e.g., horns, sirens, noisy two-wheelers, deliveries, garbage collection). In France, the limitations of using time-averaged noise metrics are at the heart of discussions within the National Noise Council (CNB) on railway and aircraft noise. The DEBATS study [10], which included an individual survey with physiological and noise measurements at home, showed that event-based acoustic indicators were more significantly associated with sleep disturbances than time-averaged energy indicators (e.g., L_n_). However, there is currently no consensus on which event-based indicators to use, and no health or regulatory recommendations refer to event-based indicators. Furthermore, these event-based indicators are not available on a large scale, unlike the L_n_ indicators which are available through strategic noise maps produced under the European Environmental Noise Directive 2002/49/EC [36]. This situation should improve in the future with the widespread use of noise maps based on event indicators.

Modelling is also challenging in dense urban centers where traffic speeds are uncertain due to frequent accelerations/decelerations from traffic lights or congestion. Moreover, noise exposure was estimated at building facades without accounting for housing sound insulation, which may limit the assessment of actual exposure, especially in areas with reinforced acoustic insulation. The deprivation index was used as a proxy for housing quality, potentially and indirectly capturing part of this effect.

Finally, recreational noise was estimated by systematically modelling a terrace for each identified establishment, whether or not one existed in reality, and using standard geometry. Despite these limitations, the recreational noise proxy is applicable at a large scale and fully replicable. Finally, individual-level factors potentially influencing the noise–sleep relationship, such as noise sensitivity [71,72,73], as well as bedroom orientation, noise adaptation behaviours (e.g., use of earplugs), and window opening [32], were not considered, potentially contributing to residual confounding.

## 7. Conclusions

Noise exposure in the Paris Metropolitan Area (~10.5 million inhabitants) is characterized by a high prevalence of inhabitants exposed to levels exceeding WHO guidelines and French legal limits, particularly for road traffic noise, which affects the largest number of people (75.6% of the population exposed above the night-time WHO guidelines of 45 dB(A), i.e., ~8.0 million inhabitants). Statistically significant positive relations were found between night-time noise exposure (AEI L_n_) and the prevalence of patients reimbursed for hypnotic psychotropic drugs, used as a proxy for chronic insomnia-type sleep disorders. Among various noise sources, road traffic noise showed the strongest association with this prevalence, followed by recreational, aircraft, and rail noise. The association between combined environmental noise and hypnotic psychotropic drug dispensing increased with age (particularly in individuals over 50), was more pronounced in women than in men, and was greater in the most socioeconomically deprived areas. Regarding public health, these findings support a more proactive public policy to prevent noise from transportation and recreational activities, aiming to prevent sleep disorders. While aircraft noise is regulated and subject to fines, cars, motorcycles, and outdoor cafés, bars and restaurants are, in our opinion, insufficiently incentivized to reduce evening and night-time noise levels.

## Figures and Tables

**Figure 1 ijerph-22-01647-f001:**
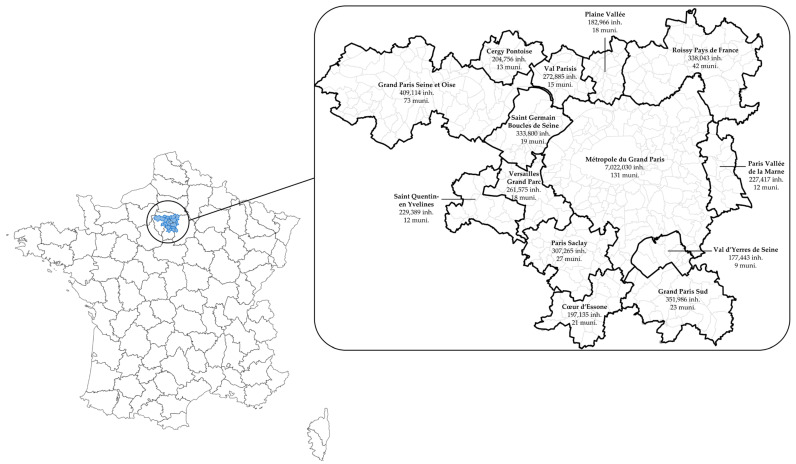
Study area. *Abbreviations*: inh., inhabitants; muni., municipalities.

**Figure 2 ijerph-22-01647-f002:**
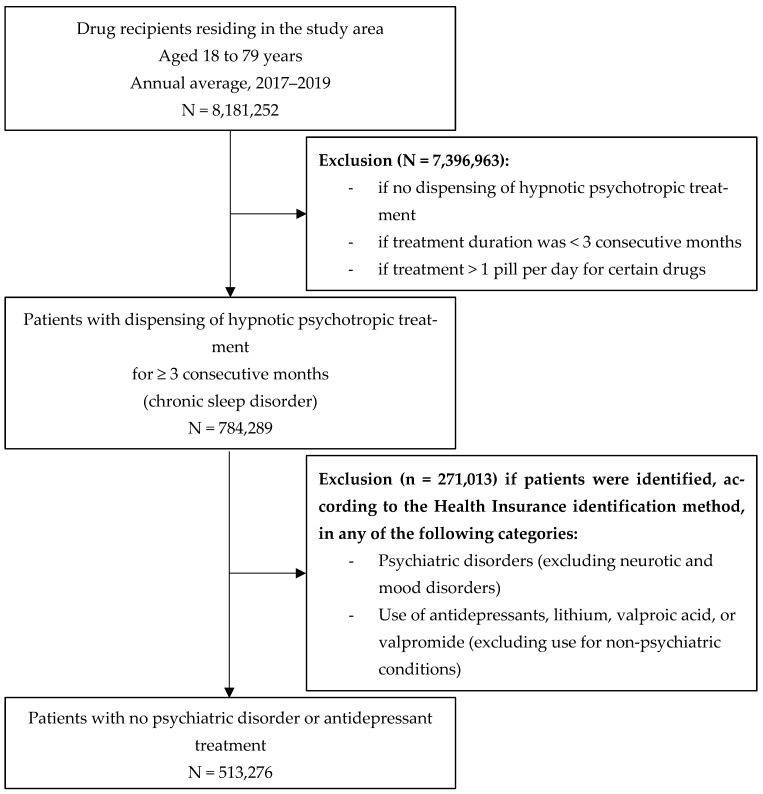
Flowchart for patient identification.

**Figure 3 ijerph-22-01647-f003:**
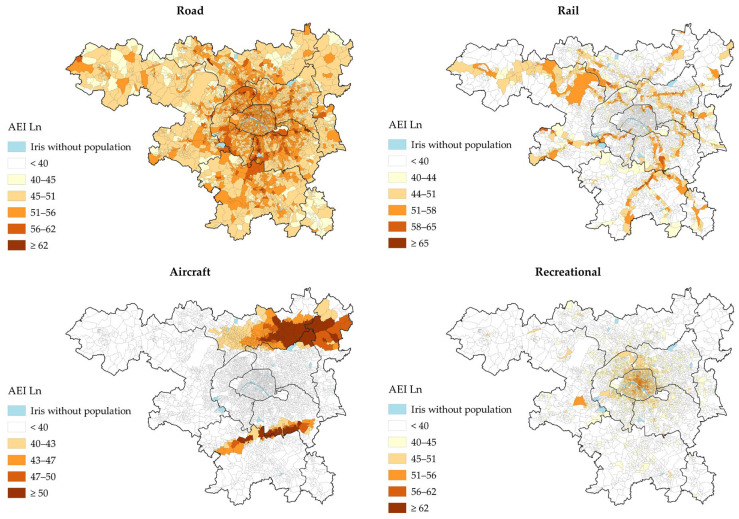
Night-time IRIS-level noise maps (AEI L_n_), in dB(A), for each environmental noise source.

**Figure 4 ijerph-22-01647-f004:**
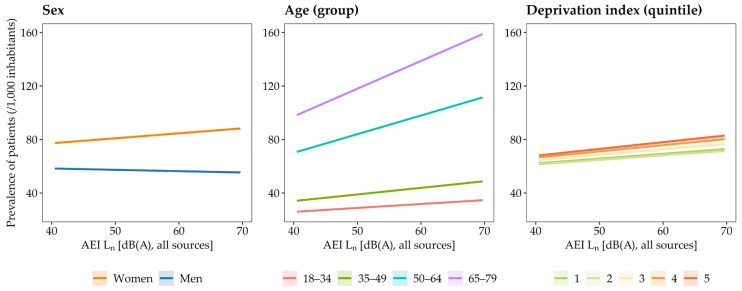
Relationships between night-time environmental noise levels (AEI L_n_) from all noise sources combined (road, rail, aircraft and recreational) and the prevalence of patients reimbursed for hypnotic psychotropic drugs, stratified by sex (**left**), age groups (**middle**), and IRIS-level deprivation index in quintiles (**right**). *Note*: quintile 1 = least disadvantaged IRIS; quintile 5 = most disadvantaged IRIS.

**Table 1 ijerph-22-01647-t001:** Prevalence of patients reimbursed for hypnotic psychotropic drugs (2017–2019) for chronic insomnia, by sociodemographic and economic factors.

	N (‰ ^1^)	*p*-Value
**Sex**		
Men (ref.)	287,987 (48.4)	-
Women	481,955 (77.0)	<0.001
**Age (years)**		
18–34 (ref.)	104,426 (26.3)	-
35–49	191,981 (53.5)	<0.001
50–64	257,299 (88.3)	<0.001
65–79	216,236 (124.8)	<0.001
**Deprivation index of the place of residence (quintile)**		
1 (ref.)	142,650 (58.4)	-
2	140,147 (59.7)	<0.001
3	154,084 (64.6)	<0.001
4	171,003 (66.0)	<0.001
5	162,058 (66.4)	<0.001

^1^ Number of patients per 1000 inhabitants. *Notes*: Patients may contribute to multiple cases, as they were identified using 3-month sliding periods to capture treatment renewal. Quintile 1 = least disadvantaged; Quintile 5: most disadvantaged.

**Table 2 ijerph-22-01647-t002:** Number and proportion of individuals exposed to night-time (L_n_) environmental noise levels according to the World Health Organization (WHO) guidelines and legal limit values set by France.

Source	<WHO	[WHO—French Legal Limit]	≥WHO	≥French Legal Limit
**Road**	2,557,949 (24.3%)	7,555,153 (71.8%)	7,957,854 (75.7%)	402,701 (3.8%)
**Rail**	9,291,472 (88.4%)	1,189,060 (11.3%)	1,224,332 (11.6%)	35,271 (0.3%)
**Aircraft**	9,488,217 (90.2%)	838,537 (8.0%)	1,029,339 (9.8%)	189,049 (1.8%)
**Recreational ^1^**	9,307,710 (88.5%)	1,181,470 (11.2%)	1,208,093 (11.5%)	26,623 (0.3%)
**Recreational ^2^**	8,691,001 (82.6%)	1,096,528 (10.4%)	1,824,802 (17.4%)	728,274 (6.93%)

*Notes*: WHO recommendations: road noise, 45 dB(A); rail noise, 44 dB(A); aircraft noise, 40 dB(A). French legal limits: road noise, 62 dB(A); rail noise, 65 dB(A); aircraft noise, 50 dB(A). ^1^ In the absence of WHO recommendations and legal limit values for recreational noise in France, those for road traffic noise were applied. ^2^ Less conservative night-time noise level, i.e., 40 dB(A) for WHO and 50 dB(A) for French legal limit.

**Table 3 ijerph-22-01647-t003:** Change (with 95% CIs) in prevalence of patients (/1000 inhabitants) reimbursed for hypnotic psychotropic drugs per 5 dB(A) increase in night-time noise level (AEI L_n_).

	Estimate (β)	95% CI	*p*-Value
Road	1.01	1.00–1.02	<0.001
Aircraft	0.51	0.50–0.51	<0.001
Rail	0.35	0.30–0.40	<0.001
Recreational	0.78	0.77–0.80	<0.001
Transportation ^1^	1.04	1.01–1.07	<0.001
Total ^2^	1.06	1.03–1.09	<0.001

^1^ Road, rail, aircraft; ^2^ Road, rail, aircraft, recreational. *Notes*: Models included each noise source with the following confounders: deprivation index quintile (categorical variable), proportion of the population reporting a primary care physician, and log-transformed population density. *Abbreviations*: CI, confidence intervals; AEI, average energetic index.

## Data Availability

The data presented in this study are available on request from the corresponding author.

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
