# Peer review of "Night-Time Exposure to Road, Railway, Aircraft, and Recreational Noise Is Associated with Hypnotic Psychotropic Drug Dispensing for Chronic Insomnia in the Paris Metropolitan Area"

_ijerph, 2025, doi:10.3390/ijerph22111647_

Round 1

Reviewer 1 Report

Comments and Suggestions for Authors

Dear authors,

The article "Night-time Exposure to Road, Railway, Aircraft, and Recreational Noise Is Associated with Hypnotic Psychotropic Drug Dispensing for Chronic Insomnia in the Paris Metropolitan Area" investigates the association between night-time exposure to various environmental noise sources and the prevalence of adults who are reimbursed for hypnotic psychotropic drugs prescribed for chronic insomnia.

Overall, the manuscript is well-written, well-designed, informative and useful. The study demonstrates a comprehensive and robust epidemiological analysis of large-scale environmental noise exposure and its association with chronic insomnia.

Amongst the findings, high prevalence of exposure to road traffic noise at night was found. Also, the studied effects were greater in people over 50 years, women, and socioeconomically deprived populations.

In my opinion this work could be published following several minor revisions.

My biggest concerns are the noise modelling, specifically the noise mapping procedure. While I understand the need to avoid overcomplicating the manuscript, more detailed information is needed regarding the noise mapping methodology. Additionally, the approach used for modelling recreational noise would benefit from being presented as bulleted information and, if possible, supplemented with schematics. As the noise mapping description is dense in places and may be difficult for readers to follow, a thorough proofreading to improve clarity and readability is recommended.

Furthermore, you mention applying a cutoff by age. According to the findings, “the impact of environmental noise on the dispensing of hypnotic psychotropic drugs increases considerably with age, especially among individuals over 50” (L270-271). It would be reasonable to assume that this effect might continue to increase with advancing age. Should there therefore be a weighting factor for age, given that the use of hypnotic psychotropic drugs is less common in younger age groups?

Please consider the following minor revisions:

L27: Please explain AEI when it appears for the first time

L89-90: An unimportant suggestion, “so-far” is redundant along with “at present”.

L103-105: Maybe there is a bias in this as well. People may use medication meant for sleep disorders for sleep and recreation (https://doi.org/10.1016/j.pbb.2021.173169). Nevertheless, I agree that this approach is more robust than self reporting methods. Also the exclusion of “addictive disorders” (L187) minimizes this bias. Also in 323-326.

L203: Please proofread this line.

L215-216: Please provide information on how you collected data on train and airport/aircraft related sources.

L213-217: Please describe or provide a reference on how you obtained the traffic flow data.

L202-221: It is not clear how you collected the data and developed the noise maps. Were the noise maps produced by you, or were they obtained from existing repositories? Please clarify what you mean by "4th term noise maps." The time period for the noise and traffic flow data should be specified, as it would likely correspond with the medical data collection period. More details are also needed on how you combined the different types of transportation noise (vehicles, trains, aircraft). Additionally, information on how you collected building height data and other early-stage noise mapping procedures would be helpful—for example, were shapefiles imported to CadnaA?

L230: Please describe NAF code for those not familiar with French legislation.

L237-237: Please explain “building footprint from the 2019 BD TOPO dataset”.

L254: Clarifications on “inverse sound propagation modelling” procedure would be helpful.

L260-263: Please explain the “correction factors” choice.

L649-652: It would be interesting to discuss the limitation of using time-averaged noise metrics given the intermittent nature of some noise sources

Kind regards

Author Response

We thank the reviewers for their thorough review of our work and for the very constructive and helpful comments provided. We have taken them into account and have reported our specific responses for each reviewer. Our responses appear below and the changes are indicated in the revised manuscript by highlighting the text in yellow.

Reviewer #1: The article "Night-time Exposure to Road, Railway, Aircraft, and Recreational Noise Is Associated with Hypnotic Psychotropic Drug Dispensing for Chronic Insomnia in the Paris Metropolitan Area" investigates the association between night-time exposure to various environmental noise sources and the prevalence of adults who are reimbursed for hypnotic psychotropic drugs prescribed for chronic insomnia.

Overall, the manuscript is well-written, well-designed, informative and useful. The study demonstrates a comprehensive and robust epidemiological analysis of large-scale environmental noise exposure and its association with chronic insomnia.

Amongst the findings, high prevalence of exposure to road traffic noise at night was found. Also, the studied effects were greater in people over 50 years, women, and socioeconomically deprived populations.

In my opinion this work could be published following several minor revisions.

My biggest concerns are the noise modelling, specifically the noise mapping procedure. While I understand the need to avoid overcomplicating the manuscript, more detailed information is needed regarding the noise mapping methodology. Additionally, the approach used for modelling recreational noise would benefit from being presented as bulleted information and, if possible, supplemented with schematics. As the noise mapping description is dense in places and may be difficult for readers to follow, a thorough proofreading to improve clarity and readability is recommended.

We thank the reviewer for the opportunity to provide additional details on the noise mapping procedure. The revised manuscript now includes further information on both transportation and recreational noise modelling (lines 207-339, pages 7-10). Moreover, supplementary figures (S1 and S2) have been added to illustrate the modelling process and to provide a clearer overview of our methodology for modelling transportation and recreational noise.

Furthermore, you mention applying a cutoff by age. According to the findings, “the impact of environmental noise on the dispensing of hypnotic psychotropic drugs increases considerably with age, especially among individuals over 50” (L270-271). It would be reasonable to assume that this effect might continue to increase with advancing age. Should there therefore be a weighting factor for age, given that the use of hypnotic psychotropic drugs is less common in younger age groups?

We agree with the reviewer that the results of our model suggest that the impact of environmental noise on the dispensing of hypnotic psychotropic drugs increases with each advancing age group. We also observed that this effect becomes more pronounced from the 50–64 age group onward, indicating that older adults (aged 50–64 and 65–79) are more at risk of chronic sleep disturbances due to environmental noise exposure. We have revised the sentence to avoid any confusion regarding a potential age cutoff (lines 699-702, page 20):

“We observed that the impact of environmental noise on the dispensing of hypnotic psychotropic drugs increases considerably with age. In particular, this increase becomes more pronounced from the 50–64 and 65–79 age groups onward compared to younger groups.”

We also agree that the use of hypnotic psychotropic drugs is less common in younger age groups, as reported in the result section “4.1 Prevalence of Chronic Dispensing of Hypnotic Psychotropic Drugs”. As reported in the statistical analysis section (lines 428-439, page 13), our first and second model included the age-standardized prevalence of patients reimbursed for hypnotic psychotropic drugs as the outcome variable to adjust for age, which may vary across IRIS units. This age-standardization corresponded to the ratio of the observed number of patients in each IRIS to the expected number if the IRIS had the same age structure as the standard population.

Please consider the following minor revisions:

L27: Please explain AEI when it appears for the first time

Done (lines 23-25, page 1).

L89-90: An unimportant suggestion, “so-far” is redundant along with “at present”.

We thank the reviewer for pointing out this redundancy. “So far” has been removed.

L103-105: Maybe there is a bias in this as well. People may use medication meant for sleep disorders for sleep and recreation (https://doi.org/10.1016/j.pbb.2021.173169). Nevertheless, I agree that this approach is more robust than self reporting methods. Also the exclusion of “addictive disorders” (L187) minimizes this bias. Also in 323-326.

We agree with the reviewer that some patients with insomnia may use these treatments for reasons other than their sleep disorders. We thank the reviewer for providing the reference and have added the following sentence, along with the associated reference, to the limitations section (lines 770-775, page 22):

“Another methodological limitation is that some patients may use these treatments for other reasons or in combination with their sleep disorders, for instance for recreational purposes [69]. We do not have data on this, but we believe that this objective measure-ment provides a more accurate estimate than simple questionnaires of insomnia preva-lence and its association with the sound environment.”

Lopez-Quintero, C.; Warren, T.; Falise, A.; Sharma, V.; Bares, C.; Oshri, A. Prevalence and Drug Use Correlates of Extra-Medical Use of Prescription Medications for Sleep among Adults in the United States: Results of the 2015-2018 National Survey on Drug Use and Health. Pharmacol Biochem Behav 2021, 204, 173169, doi:10.1016/j.pbb.2021.173169.

L203: Please proofread this line.

We have revised this sentence as follows (lines 207-212, page 7):

“Noise levels from road, rail and aircraft traffic were estimated using the 4th round noise maps produced and published in 2023 under the framework of the European Environmental Noise Directive (2002/49/EC) [43], which requires public authorities to produce noise maps every 5 years for large urban areas (≥100,000 inhabitants) and ma-jor transport infrastructures.”

L215-216: Please provide information on how you collected data on train and airport/aircraft related sources.

We thank the reviewer for pointing out this lack of information and for the opportunity to provide additional details on the methodology of noise modelling and the data sources. Data on train and airport/aircraft sources from various stakeholders, including local authorities, infrastructure managers, and flat-rate data. They incorporated the following contributions:

  • CEREMA, for road and rail traffic in areas located outside the Métropole du Grand Paris, covering road infrastructure with >3 million vehicles per year and railways with >30,000 trains per year;
  • ADP (Paris Airports) and the DGAC (French Civil Aviation Authority), for aircraft traffic at major airports (>50,000 movements per year).

The revised version of the manuscript now contains this information (lines 230-267, page 8).

L213-217: Please describe or provide a reference on how you obtained the traffic flow data.

As stated in our previous response, traffic flow data were obtained from various stakeholders, including local authorities, infrastructure managers, and flat-rate data.

L202-221: It is not clear how you collected the data and developed the noise maps. Were the noise maps produced by you, or were they obtained from existing repositories? Please clarify what you mean by "4th term noise maps." The time period for the noise and traffic flow data should be specified, as it would likely correspond with the medical data collection period. More details are also needed on how you combined the different types of transportation noise (vehicles, trains, aircraft). Additionally, information on how you collected building height data and other early-stage noise mapping procedures would be helpful—for example, were shapefiles imported to CadnaA?

We are aware that we have failed to provide sufficient information. The revised version of the manuscript now contains additional details on data and developed the noise maps (lines 207-267, pages 7-8). Bruitparif, as a co-author of the manuscript, produced noise maps incorporating contributions from CEREMA, ADP, and DGAC for major transport infrastructures (road, rail, and aircraft). These contributions were complemented by Bruitparif across the study area through modelling transportation noise source emissions and propagation using CadnaA software, with input data obtained from various stakeholders, including local authorities, infrastructure managers, and flat-rate data. Transportation noise emissions were calculated from average annual traffic conditions, including traffic flow, speeds, traffic composition (light vehicles, heavy goods vehicles, motorized two-wheelers), road surface type, types of trains, railway track equipment, average flight paths, and aircraft fleet composition. For road noise, traffic data were derived from traffic counts available in the area (permanent counters, temporary campaigns, cameras, data from navigation applications, etc.). For railway noise, the input data (traffic, train types, infrastructure characteristics) came exclusively from infrastructure managers (RATP and SNCF). For aircraft noise, calculations were per-formed by airport management services for air traffic control authorities (DGAC) based on annual averages in terms of flight paths and number of movements per aircraft type.

L230: Please describe NAF code for those not familiar with French legislation.

Done (lines 283-285, page 9):

“A filter was applied to retain only food service activities (code 56.10A) and beverage-serving activities (code 56.30Z) according to the French classification of activities [48], and duplicate entries were removed.”

L237-237: Please explain “building footprint from the 2019 BD TOPO dataset”.

The virtual terrace surfaces were adjusted to the geometry of the nearest building façade referenced in the 2019 BD TOPO®. Corrected lines 299-300, page 10.

L254: Clarifications on “inverse sound propagation modelling” procedure would be helpful.

We have provided the following clarification (lines 316-319, page 10):

“This methodology, implemented using CadnaA software, involves deducing the sound power of emitting surfaces, defined by their geometry and surface area, from actual measurements taken at several receiving points. The software then automatically adjusts the source levels until the simulation best matches the measurements.”

L260-263: Please explain the “correction factors” choice.

Done (lines 321-333, page 10):

“In the present study, the acoustic power levels were adjusted based on the ratio between the total terrace surface area in the Parisian neighbourhood of Halles-Beaubourg-Montorgueil (10 288 m²) and the total terrace surface area in the present study (30 763 m²). This resulted in a correction factor of −5 dB for the acoustic power levels emitted by terraces, approximating the logarithmic ratio of the total terrace areas obtained using the two identification methods, calculated as . This correction involved adjusting the acoustic power levels emitted by terraces to ensure that the estimated recreational noise impact on the population remained consistent with that of the Halles-Beaubourg-Montorgueil Parisian neighbourhood [49]. An additional correction of -3 dB was applied to adjust from an assessment based on the summer period to an annual average assessment. This correction amounts to considering that terraces are in operation for about half of the year.”

L649-652: It would be interesting to discuss the limitation of using time-averaged noise metrics given the intermittent nature of some noise sources

We agree with the reviewer that using time-averaged noise metrics is a limitation when assessing the intermittent nature of certain noise sources. The limitations section now includes the following discussion (lines 787-800, page 22):

“Additionally, the Ln indicator from noise maps reflects average exposure and cannot capture temporal patterns or specific intermittent events, particularly from air and rail traffic, and certain road traffic sources (e.g., horns, sirens, noisy two-wheelers, deliver-ies, garbage collection). In France, the limitations of using time-averaged noise metrics are at the heart of discussions within the National Noise Council (CNB) on railway and aircraft noise. The DEBATS study [10], which included an individual survey with phys-iological and noise measurements at home, showed that event-based acoustic indicators were more significantly associated with sleep disturbances than time-averaged energy indicators (e.g., Ln). However, there is currently no consensus on which event-based in-dicators to use, and no health or regulatory recommendations refer to event-based in-dicators. Furthermore, these event-based indicators are not available on a large scale, unlike the Ln indicator which are available through strategic noise maps produced un-der the European Environmental Noise Directive 2002/49/EC [43]. This situation should improve in the future with the widespread use of noise maps based on event indicators.”

Reviewer 2 Report

Comments and Suggestions for Authors

"Night-time Exposure to Road, Railway, Aircraft, and Recreational Noise Is Associated with Hypnotic Psychotropic Drug Dispensing for Chronic Insomnia in the Paris Metropolitan Area" is a interesting and important paper, but it used only data of the Paris Metropolitan Area

= big limitation

Comments on the Quality of English Language

the linguistic style should be improved by a native speaker

Author Response

We thank the reviewers for their thorough review of our work and for the very constructive and helpful comments provided. We have taken them into account and have reported our specific responses for each reviewer. Our responses appear below and the changes are indicated in the revised manuscript by highlighting the text in yellow.

Reviewer #2: "Night-time Exposure to Road, Railway, Aircraft, and Recreational Noise Is Associated with Hypnotic Psychotropic Drug Dispensing for Chronic Insomnia in the Paris Metropolitan Area" is a interesting and important paper, but it used only data of the Paris Metropolitan Area

= big limitation

We agree with the reviewer. Indeed, the study is limited to the Paris region, but we believe that the lifestyle in this metropolis is similar to that of many other cities worldwide, and that our observations may be of interest to health authorities in those cities. Furthermore, we propose a recreational noise proxy that can be applied at a large scale and is fully replicable.

Reviewer 3 Report

Comments and Suggestions for Authors

Thanks for all your work on this. It looks good to me and I have mostly quite minor comments and suggestions for you as follow:

L27: Given that this is the Abstract, the text “Each 5 dB(A) increase in AEI Ln . . .” needs some more explanation to be meaningful to readers.

L90: the words “so far” seem redundant in the sentence because it already has “at present” earlier.

L92: perhaps specify “systemic health” here. There have been studies into the effects of recreational noise on noise-induced hearing loss, which I understand you are not wanting to get into.

L111: “factor” should be “factors”

L151 and onwards: I don’t recognise the term “specialties” in this context and wonder if it may be a translation problem: in English, one can describe a drug as a “specific” though that might be a bit archaic. Could it just say “pharmaceuticals” or "drugs"? I should say that I am not an expert in pharmaceutics, so it may simply be my lack of knowledge.

L172: “patients’” should just be “patient”

Figure 2: The second box is too small so some of the text is missing.

L287: I understand the lack of guidelines for recreational noise, and using the level for road noise seems as good as any. I presume you would have put thought into the decision, though. Do you have more of a rationale? Did you run sensitivity analyses at other levels?

L300: there is an extra “period” here.

L301: I don’t understand why it says “(the difference between 𝐿𝑛j and 𝐿𝑛j+1 is equal to 1 dB[A])”. Are you just pointing our that Lnj is measured in dBA?

L306: I suggest removing the phrase “by traffic noise” from this sentence because the term “traffic” is often used colloquially to refer to road traffic, and I was momentarily confused when I first read this.

Results generally: Quite a lot of the text includes statistics that are also included in the tables. I prefer the approach of referring to key points in the tables and explaining the meaning intuitively without repeating the statistical information and values, since it renders the text more readable and avoids repetition.

Table 1: “Sex” is spelled as in French.

Table 2: This is currently a bit hard to read. I have given it some thought, and can see the difficulty you had, so I am not insisting that you change it, but rather suggest that you consider putting the WHO and French limits into the footnotes to clear out the table a bit. See what you think and however you decide is fine with me.

Figure 4: There appears to be an interaction in that women’s use of drugs increases with increasing noise exposure whereas men’s does not, or possibly even decreases. Is this preserved across age groups and deprivation levels? If so, given that we know men are vulnerable to noise, does this imply that the method you have used is not applicable to men?

L441: Saying “originally” here implies that you then did something else. I suggest removing the word.

 L444: I do think your results are interesting, but still suggest you take out the word “interestingly” here since this is scientific writing and it seems inappropriate.

L455: this should specify “road” traffic.

L604-608: as I mentioned above, if you investigated interactions involving sex and noise levels with the other demographic variables, you could comment on this more.

Author Response

We thank the reviewers for their thorough review of our work and for the very constructive and helpful comments provided. We have taken them into account and have reported our specific responses for each reviewer. Our responses appear below and the changes are indicated in the revised manuscript by highlighting the text in yellow.

Reviewer #3: Thanks for all your work on this. It looks good to me and I have mostly quite minor comments and suggestions for you as follow:

L27: Given that this is the Abstract, the text “Each 5 dB(A) increase in AEI Ln . . .” needs some more explanation to be meaningful to readers.

We have provided detailed information on “AEI Ln” to make it more meaningful to readers (lines 18-25, page 1):

“We addressed this gap by conducting a large-scale ecological study across the Paris Metropolitan Area (~10.5 million inhabitants) examining associations between the Av-erage Energetic Index of night-time noise (AEI Ln) from road, aircraft, railway, and rec-reational sources and the prevalence of adults aged 18–79 reimbursed for hypnotic psy-chotropic drugs prescribed for chronic insomnia between 2017 and 2019, stratified by sex, age, and socioeconomic status. The AEI Ln represents the population-weighted av-erage energy noise level within each territory at night (22:00–06:00 in France), calculat-ed at the IRIS level (~2,487 inhabitants per IRIS).”

L90: the words “so far” seem redundant in the sentence because it already has “at present” earlier.

We thank the reviewer for pointing out this redundancy. “So far” has been removed.

L92: perhaps specify “systemic health” here. There have been studies into the effects of recreational noise on noise-induced hearing loss, which I understand you are not wanting to get into.

We agree with the reviewer. We have specified (lines 94-97, page 2):

“To the best of our knowledge, only one study has examined noise from recreational activities in relation to non-auditory health outcomes, finding a significant association with increased hypertension risk, but without assessing its effects on sleep disturbances [27].”

L111: “factor” should be “factors”

Done. Thanks.

L151 and onwards: I don’t recognise the term “specialties” in this context and wonder if it may be a translation problem: in English, one can describe a drug as a “specific” though that might be a bit archaic. Could it just say “pharmaceuticals” or "drugs"? I should say that I am not an expert in pharmaceutics, so it may simply be my lack of knowledge.

We apologize for the confusion. We have replaced the term “specialties” by the term “drug” throughout the revised version of the manuscript.

L172: “patients’” should just be “patient”

Done. Thanks.

Figure 2: The second box is too small so some of the text is missing.

We apologize for the missing text. We have now included the complete flowchart.

L287: I understand the lack of guidelines for recreational noise, and using the level for road noise seems as good as any. I presume you would have put thought into the decision, though. Do you have more of a rationale? Did you run sensitivity analyses at other levels?

Indeed, in the absence of recommendations for recreational noise, those for road traffic noise were applied. The justification has been added in the revised version of the manuscript. As requested, we performed a sensitivity analysis on the number and proportion of individuals exposed to different levels of recreational noise. We provided the prevalence of exposure to night-time recreational noise levels ≥40 dB(A) in Table 2, as suggested by the 2009 WHO Night Noise Guidelines as the lowest observed adverse effect level for night noise, associated with self-reported sleep disturbance, environmental insomnia, and increased use of somnifacient drugs and sedatives (also justified in lines 361-369, page 11):

“In the absence of WHO recommendations and legal limit values for recreational noise in France, those for road traffic noise were applied as the most conservative benchmark. This choice is also justified as recreational noise is temporally more similar to road noise than to railway or aircraft noise, the latter being event-driven and producing roughly regular peaks. The population exposed to a less conservative night-time noise level, i.e., 40 dB(A) and 50 dB(A), was also estimated for recreational noise. The 40 dB(A) threshold has been suggested as the lowest observed adverse effect level for night noise, associated with self-reported sleep disturbance, environmental insomnia, and increased use of somnifacient drugs and sedatives [3,50].”

World Health Organization (WHO) Night Noise Guidelines for Europe; World Health Organization. Regional Office for Europe, 2009.

L300: there is an extra “period” here.

Corrected.

L301: I don’t understand why it says “(the difference between ??j and ??j+1 is equal to 1 dB[A])”. Are you just pointing our that Lnj is measured in dBA?

This is the “resolution” of noise levels: 1 dB(A) interval.

L306: I suggest removing the phrase “by traffic noise” from this sentence because the term “traffic” is often used colloquially to refer to road traffic, and I was momentarily confused when I first read this.

We agree. The term “by traffic noise” has been removed.

Results generally: Quite a lot of the text includes statistics that are also included in the tables. I prefer the approach of referring to key points in the tables and explaining the meaning intuitively without repeating the statistical information and values, since it renders the text more readable and avoids repetition.

We agree. We have removed statistical information and repeated values from the text when they are available in the tables.

Table 1: “Sex” is spelled as in French.

Corrected. Thanks.

Table 2: This is currently a bit hard to read. I have given it some thought, and can see the difficulty you had, so I am not insisting that you change it, but rather suggest that you consider putting the WHO and French limits into the footnotes to clear out the table a bit. See what you think and however you decide is fine with me.

Thank you for your advice on clarifying the table. We have added the WHO and French limits into the footnotes.

Figure 4: There appears to be an interaction in that women’s use of drugs increases with increasing noise exposure whereas men’s does not, or possibly even decreases. Is this preserved across age groups and deprivation levels? If so, given that we know men are vulnerable to noise, does this imply that the method you have used is not applicable to men?

We thank you for this relevant comment. We performed additional analyses including a three-way interaction to explore whether the effect of noise exposure on the prevalence of patients reimbursed for hypnotic psychotropic drugs is consistent across age groups and deprivation levels. We observed that the effect of noise exposure increases across all age groups in women. In men, it increases only from the 50–64 and 65–79 age groups. Similarly, we observed that the effect of noise exposure increased across all deprivation index quintiles among women, whereas it increased only from the most deprived quintiles (4 and 5) among men. These results reinforce the idea that women are more vulnerable to noise than men at all ages and socio-economic levels. Men are also affected, but only from the 50–64 and 65–79 age groups onward and in the most socioeconomically deprived areas. These results have been added in supplementary materiel (Table S2). The statistical methodology has been added lines 443-449, page 13.

L441: Saying “originally” here implies that you then did something else. I suggest removing the word.

We agree. The term “originally” has been removed.

 L444: I do think your results are interesting, but still suggest you take out the word “interestingly” here since this is scientific writing and it seems inappropriate.

We agree. The term “interestingly” has been removed.

L455: this should specify “road” traffic.

Done.

L604-608: as I mentioned above, if you investigated interactions involving sex and noise levels with the other demographic variables, you could comment on this more.

When performing a three-way interaction model, we observed results that contradicted those of Roswall et al. (2020). We found significant positive associations between noise exposure and the prescription of medications in both men and women aged 50-64, whereas they reported significant associations only in men, but not in women. Their study focused solely on road traffic noise, which makes direct comparisons with ours difficult. We have added further details in the Discussion section regarding our additional analyses on the three-way interactions (lines 697-763, pages 20-22).